# Medial packing and elastic asymmetry stabilize the double-gyroid in block copolymers

Abhiram Reddy[1], Michael S. Dimitriyev [1] & Gregory M. Grason [1✉]

Triply-periodic networks are among the most complex and functionally valuable self-assembled morphologies, yet they form in nearly every class of biological and synthetic soft matter building blocks. In contrast to simpler assembly motifs – spheres, cylinders, layers – networks require molecules to occupy variable local environments, confounding attempts to understand their formation. Here, we examine the double-gyroid network phase by using a geometric formulation of the strong stretching theory of block copolymer melts, a prototypical soft self-assembly system. The theory establishes the direct link between molecular packing, assembly thermodynamics and the medial map, a generic measure of the geometric center of complex shapes. We show that "medial packing" is essential for stability of double-gyroid in strongly-segregated melts, reconciling a long-standing contradiction between infinite- and finite-segregation theories. Additionally, we find a previously unrecognized non-monotonic dependence of network stability on the relative entropic elastic stiffness of matrix-forming to tubular-network forming blocks. The composition window of stable double-gyroid widens for both large and small elastic asymmetry, contradicting intuitive notions that packing frustration is localized to the tubular domains. This study demonstrates the utility of optimized medial tessellations for understanding soft-molecular assembly and packing frustration via an approach that is readily generalizable far beyond gyroids in neat block copolymers.

[1] Department of Polymer Science and Engineering, University of Massachusetts, Amherst, MA 01003, USA. ✉email: grason@umass.edu

Triply-periodic network morphologies constitute "natural forms" of self-assembled soft matter. Forming in nearly every class of amphiphilic molecular building blocks, from surfactants[1–3] and lyotropic liquid crystals[4] to complex shape amphiphiles[5,6], network domain structures most typically occur under conditions intermediate to those where cylindrical and layered morphologies, two other canonical forms of soft matter organization[7], form. By far the most commonly observed network structures are the cubic domain networks: the double-gyroid (DG), double-diamond (DD) and double-primitive (DP). Heuristically, the local structure of these "in-between" phases is often characterized by the shape of the intermaterial dividing surface (IMDS) that separates unlike components[8], which is effectively more curved than planar layers, but less-so than cylinders. Likewise, their global structure reflects a hybridization between cylinders and layers[9], with one domain forming a double network of interconnected tubes (e.g. meeting at 3-, 4-, and 6-valent connections for DG, DD, and DP respectively), interspersed with a slab-like matrix layer, whose undulating shape approximates a triply-periodic minimal surface (e.g. the Gyroid, Diamond, Primitive surfaces). The combination of intercatenated, polycontinuous domain structures with complex (3D crystallographic) symmetries make triply-periodic network morphologies among the most sought after self-assembly architectures for functional nanostructured materials[10,11] and their robust formation pathways make them ideal naturally-occurring photonic structures, endowing organisms from butterflies and beetles to birds with structural coloration[12,13].

Despite their formation by diverse molecular systems, universal explanations for why various network structures form under equilibrium assembly remain elusive. A common theme in most rationalizations of network formation is the concept of *packing frustration*[14–18], which loosely refers to the incompatibility between a preferred local geometry for constituents and constraints of space filling. Unlike cylinders or layers, which have uniform shape and thickness, IMDSs of tubular networks have variable curvature and molecules extending from this surface to the "center" of the domain have to reach variable distances, yet need to do so at nearly constant density. While this heuristic picture is widely held, a basic gap remains between specific geometric measures of packing and IMDS shape, molecular configurations of constituents, and the thermodynamic selection of network structures that form in equilibrium.

In this paper, we revisit the theory of self-assembly of network morphologies based on the so-called *strong-segregation theory* (SST) of AB block copolymer melts. Block copolymer melt assembly is an ideal system for understanding complex morphology formation, owing to the fact that equilibrium behavior of variable chemical and architectural compositions are well captured by a self-consistent field (SCF) model[19], dependent on only a few parameters: the composition fractions $f$ of component blocks; the $\chi$ parameter describing immiscibility between unlike components; and chain length $N$ ($\chi N$ controls the effective degree of segregation in assembled morphologies). Nonetheless, the understanding of the stable network phases of block copolymer melts has a complex and unsettled history[20].

While initial experimental observations, based on AB starblocks, pointed to a DD (symmetry group $Pn\bar{3}m$) network[21], subsequent studies of linear AB diblocks[22] identified the DG ($Ia\bar{3}d$) as the stable network phase (see Fig. 1a) at compositions intermediate to stable hexagonal cylinders (Hex) and lamellar (Lam) morphologies. This was confirmed by computational SCF studies of triply-periodic morphologies[23], showing that DG was the equilibrium phase between Hex and Lam phases at low-to-moderate $\chi N$. Extrapolations of phase boundaries to higher segregation first suggested that the stability window of DG would

pinch off at sufficiently high $\chi N$[24]. A parallel set of studies considered the stable network morphologies in the asymptotic limit $\chi N \to \infty$ based on SST[25–28], which computes the free energies directly from space-filling configurations of alternating, brush-like domains of block copolymers in competing arrangements. Within SST, the thermodynamics become almost a purely geometric balance between entropic costs of chain extension and enthalpic cost proportional to the IMDS area[16]. From this perspective, those stable phases in the SST limit might be viewed as "natural forms" of assembly, emerging from generic, geometric considerations. Early SST calculations predicted that DG had the lowest free energy among competitor networks, but was nevertheless not stable relative to Hex and Lam as $\chi N \to \infty$[26,28], excepting for experimentally extreme conditions where matrix-forming blocks are nearly an order of magnitude more stiff than tubular blocks. However, more recent experiments[29] and SCF calculations[30] that push to much higher $\chi N$ show that equilibrium DGs, for equal or nearly symmetric block stiffness, nevertheless persist to very high degrees of segregation. Whether DG belongs to those natural forms stable in the $\chi N \to \infty$ limit, or otherwise relies on specific entropic corrections at finite segregation, has remained an open question[31].

Here, we show that the resolution of this long-standing puzzle derives from a close connection between packing frustration of constituent chains and *medial geometry* of complex IMDS shapes. Specifically, we generalize the SST approach to network morphologies to incorporate degrees of freedom associated with the so-called terminal boundaries[20] between brush-like domains and show that entropic considerations favor spreading of termini over the medial sets of tubular network domains. This medial packing not only leads to thermodynamic stability of the DG in the SST phase diagram, but further predicts a DG stability window that depends non-monotonically on the elastic asymmetry between blocks that compose the tubular and matrix domains. We further demonstrate the features of medial packing of subdomain chains that persist to finite degrees of segregation. While we restrict our focus to the stability of one network morphology (DG) for a particular class of soft molecular assembly (AB diblock melts), we posit that the basic paradigm and specific methodology for evaluating the thermodynamic costs of packing frustration extend to other complex morphologies assembled from a much broader class of soft matter building blocks.

## Results

**Medial anatomy of network morphologies.** To understand the thermodynamic costs of packing in network morphologies, it is necessary to measure how far chains must extend from the IMDS into the "center" of complex domains. In block copolymer melts, these centers, defined as terminal boundaries[20], are locally 2D sets of contact points between opposing brush-like domains from distinct regions of the the IMDS. The length $h$ of the molecular extension from the IMDS to the terminal boundary defines a local thickness.

Historically, the focus on packing frustration in network morphologies of neat (solvent-free) amphiphiles like block copolymers, as well as "Type I" lyotropic network phases[32], has been on the tubular, double-network domains (e.g. the red A domain in Fig. 1a). Prior theories of network thermodynamics in melts associated the centers of tubular domains with their skeletal graphs[26,28,33], 1D graphs that thread between adjacent $Z$-valent junctions, or nodes. Figure 1b shows the skeletal graph within one of the two single-gyroid domains in the DG morphology, which connects the node centers located at 8 of the 16b Wyckoff positions of $Ia\bar{3}d$. Focusing on a single 3-valent nodal region in Fig. 1c, the largest distance from the IMDS (gray) to the skeletal graph (the

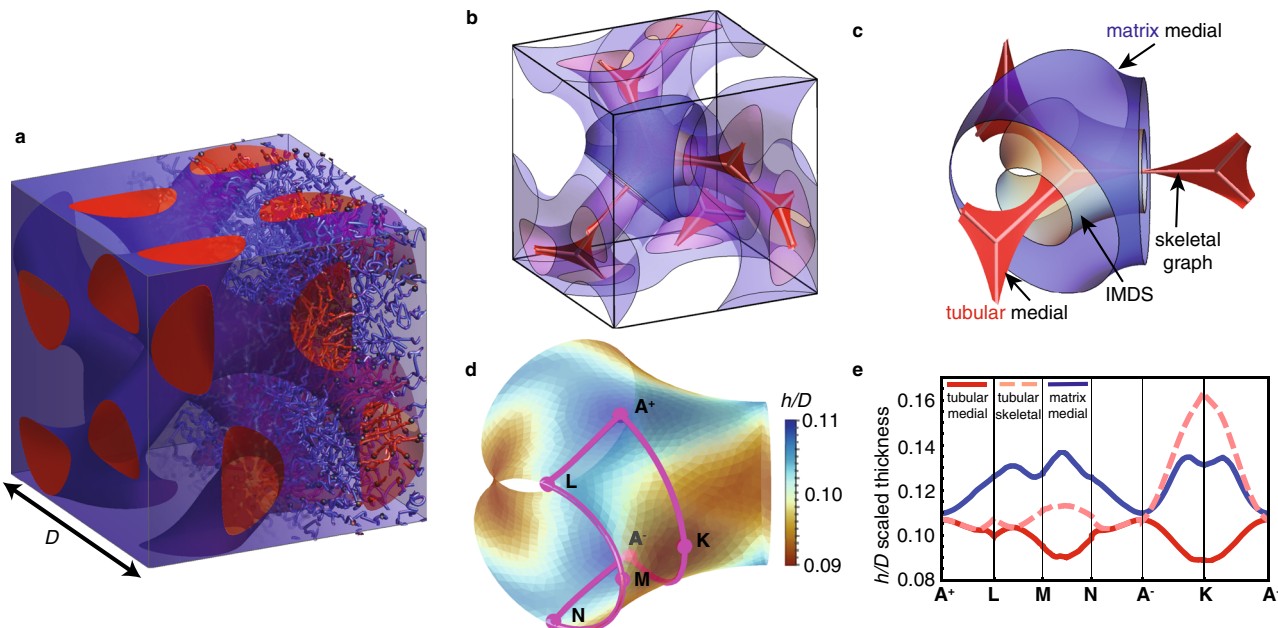

**Fig. 1 Geometric measures of local thickness in double-gyroids.** Schematic of chain packing in tubular and matrix domains of self-assembled diblock copolymer double-gyroid (DG) morphology with cubic unit cell length $D$ is shown in (**a**), where red and blue regions show domains of A and B segments. **b** Shows only a single-gyroid domain of the structure shown in (**a**) with red and blue surfaces corresponding to medial sets of single tubular and matrix domains and the skeletal graph shown as light red, with the highlighted nodal region shown in detail in (**c**). The spatial map of the tubular domain medial thickness is plotted on the nodal IMDS in (**d**), with a path bounding an asymmetric unit of the DG: 3- and twofold axes pass through points $A^{\pm}$ and **K**. **e** shows a "band diagram" of local domain thickness $h$ as measured by distance to medial surface or skeletal graph.

maximal skeletal thickness) is associated with points from the saddle-like regions between struts that have to extend towards the center of the node.

Following an argument proposed by Schröder-Turk, Fogden and Hyde for lyotropic morphologies and block copolymers[32,34,35], it was shown[20] that a more natural definition of local thickness is provided by the medial map, the set of points closest to a given point on a generating surface (or set of surfaces). Points on the generating surface are mapped to medial sets by following the local surface normal until it intersects with another, remote surface normal at an equidistant point; the distance to those points $h$ is the medial thickness[20,34]. This construction rigorously minimizes the distance map of all points enclosed within the volume bounded by the generating surface to that surface. Thus, the medial map can be viewed as a generalization of *centroidal* Voronoi cells[36], in which distances to a set of generating points are minimized, to the case of generating surfaces. Figure 1b, c also shows the medial sets for both the outer (matrix) and inner (tubular) domains for a level set model for DG. The outer medial surface, which divides the DG into two enantiomeric single-gyroid domains, closely resembles the Gyroid (G) minimal surface, while the inner medial set is composed of a web-like surface that locally spreads out in the plane of the threefold junctions, and twists by $70.5°$ from one node to the next.

Figure 1d plots a spatial map of the medial thickness of the (inner) tubular domain on the nodal region of the IMDS of the DG model. In the Fig. 1e "band diagram," we compare domain thickness measures along a closed path passing through symmetry points of the nodal region. Notably, the magnitude and variation of skeletal thickness is not only greater than medial thickness, but the spatial pattern of thickness variation is nearly opposite between these measures. While the quasi-planar, three-fold points $A^{\pm}$ correspond to minimal skeletal thickness, these are points of maximal medial thickness. This contrast is even greater at the saddle-like "elbows" at twofold points **K**, where the maximal skeletal thickness is nearly double the medial thickness, which is at a local minimum. Contrary

to the standard heuristic that the cost of packing frustration in network phases is confined to tubular domains, the medial thickness in the matrix domain (also shown in Fig. 1e) exhibits variability comparable to the tubular domain. These geometric observations strongly imply that theories based on the assumption of skeletal packing (i.e. terminal boundaries lie along the 1D skeleton) dramatically overestimate the costs of chain stretching in the tubular block relative to medial packing (i.e. terminal boundaries lie along the medial set). As medial geometry encodes the shortest distance to the "center" of a complex morphology, previous studies have proposed that medial thickness can be used as a heuristic measure of packing frustration[17,20,35] or otherwise as an ingredient in phenomenological models of its costs[37] in network morphologies. We next exploit a fully molecular description of network assembly, the SST of block copolymer melts, to test and establish the direct connections between the medial geometry of complex networks, the underlying configurations of molecular constituents and the thermodynamic stability of the DG phase.

**Thermodynamics of medial packing in double-gyroid.** To assess the importance of terminal spreading along medial sets, we turn to the SST of DG (and its competitor phases Hex and Lam)[26,27,38]. Each block copolymer consists of $N$ segments, a fraction $f$ of which are A-type, with segment lengths $a_A$ and $a_B$. As shown schematically in Fig. 2, microphase segregation partitions space into A and B domains, delineated by an IMDS, from which extended blocks form brushes up to a thickness defined by the terminal boundary. Critically, as a solvent-free melt, volumes occupied by chains must obey a local volume balance constraint: the ratio occupied by A:B blocks extending from each point on the IMDS is $f:(1-f)$. Given a volume-balanced space partition for morphology $X$, the free energy per chain (in units of $k_B T$) is

$$F(X) = \frac{N\rho^{-1}}{V(X)}\left[\gamma A(X) + \frac{\kappa_A}{2} I_A(X) + \frac{\kappa_B}{2} I_B(X)\right] \quad (1)$$

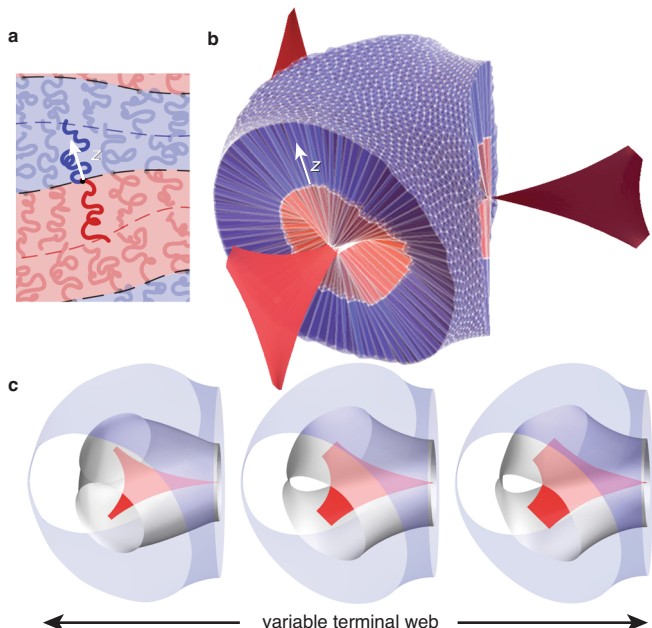

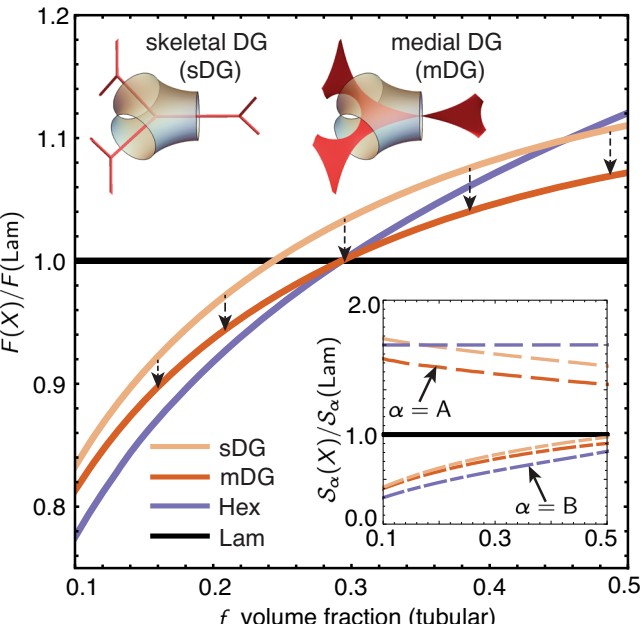

**Fig. 2 Subdomain tessellations for strong-segregation chain packing.** Schematic of chain packing within a variably-shaped morphology is shown in (**a**) where $z$ labels local extension of blocks from the IMDS. **b** Shows a (volume balanced) tessellation of the nodal region of DG, generated via the medial SST construction. **c** Shows a sequence of medial SST geometries (at fixed composition $f = 0.3$) highlighting lateral spreading of terminal webs achieved via parametric variation of DG generating surfaces (as described in the text).

**Fig. 3 Medial versus skeletal chain packing thermodynamics.** Free energy comparison of medial-DG (mDG) and skeletal-DG (sDG) to competitor hexagonal cylinder (Hex) and lamellar (Lam) phases, for conformationally symmetric linear diblocks. Inset shows scaled entropic cost of stretching for both blocks (compared at equal IMDS area per chain).

where $\rho V(X)/N$ is the number of chains in the assembly with total volume $V(X)$ and volume per segment $\rho^{-1}$ (see Supplementary Note 1). Remarkably, SST samples the relevant conformational fluctuations of underlying chains in the $\chi N \rightarrow \infty$ limit[39], and yet the free energy reduces to purely geometric measures of competing morphologies[16]. The first term in Eq. (1) encodes the surface energy of AB contact, where $A(X)$ is the total area of the IMDS and $\gamma \propto \sqrt{\chi}$ is effective surface tension. The second and third terms encode the respective entropic costs of stretching Gaussian chains[27] within the *parabolic brush* theory (PBT),

$$I_\alpha = \int_{V_\alpha} \mathrm{d}V \, z^2 \,, \text{ with } \alpha = \mathrm{A, B} \qquad (2)$$

where $z$ is the chain extension from the IMDS (see Fig. 2a). Coefficients $\kappa_\alpha$ (Supplementary Eq. (4)) describe molecular and architectural features of the blocks that control the effective entropic stiffness of the domains. While the PBT assumes free ends to be distributed throughout brush, this is violated for convex brush curvatures[40] resulting in an end-exclusion zone. We show (Supplementary Note 10) that corrections to PBT for the variable convex domain shapes involved[41] have negligible impact on the stability of network phases.

Per Eq. (2), maximizing entropy of component blocks is equivalent to minimizing second-moments of volume measured relative to the IMDS. Chain packing is, in this sense, a variant of the quantizer problem[36], which seeks tessellations that minimize the second-moment of distance from a set of generating points, generalized to measure distance with respect to generating surfaces. Given an IMDS shape, the medial map would provide the optimal $I_\alpha$ for each domain considered in isolation. However, as shown in Supplementary Note 3, the medial map generically fails to satisfy the local volume balance constraint. Moreover,

stable morphologies also optimize the competition between IMDS area minimization and block stretching.

To model this thermodynamic competition and test how closely physical assemblies come to realizing medial packing, we apply the following *medial SST* approach to DG morphologies (see Methods, Supplementary Note 3–4). In brief, our approach, summarized in Supplementary Fig. 7, uses the medial map of gyroidal surfaces to generate terminal sets (both A and B ends) from which a set of space filling chain trajectories are derived along with an IMDS consistent with the volume balance constraint for each chain trajectory, resulting in mutually compatible terminal boundaries, IMDS, and trajectories, as shown in Fig. 2b. We then optimize the computed free energies over the variational family of generating gyroidal surfaces – corresponding to variable shapes and sizes of the terminal surfaces as shown Fig. 2c—to determine and analyze thermo-dynamically preferred DG chain packing. Notably, local volume balance generally requires some measure of tilt between trajectories and the IMDS, a signature of the deviation from strictly medial packing which we return to below.

In Fig. 3 we compare the free energy of DG to its competitor phases (Lam and Hex) for conformationally symmetric (i.e. chains of equal segment lengths $a_\mathrm{A}$ and $a_\mathrm{B}$) linear AB diblocks, using a variable-IMDS shape SST for Hex[42]. We also compare the medial SST construction of DG (medial-DG) to a skeletal construction (skeletal-DG) that is based on and closely approximates Milner and Olmsted's skeletal SST[26] for network phases (see Supplementary Note 5), showing that the free energy per chain in medial packing is significantly lower ($\approx 2 - 4\%$) than skeletal packing. In the inset of Fig. 3, we plot

$$\mathcal{S}_\alpha(X) = I_\alpha(X) A^2(X) / V^3(X), \qquad (3)$$

a dimensionless measure of entropic stretching that compares block entropies at an equal IMDS area per chain (i.e. fixed interfacial cost, see Supplementary Note 2). This shows that the free energy drop of medial-DG relative to skeletal-DG primarily

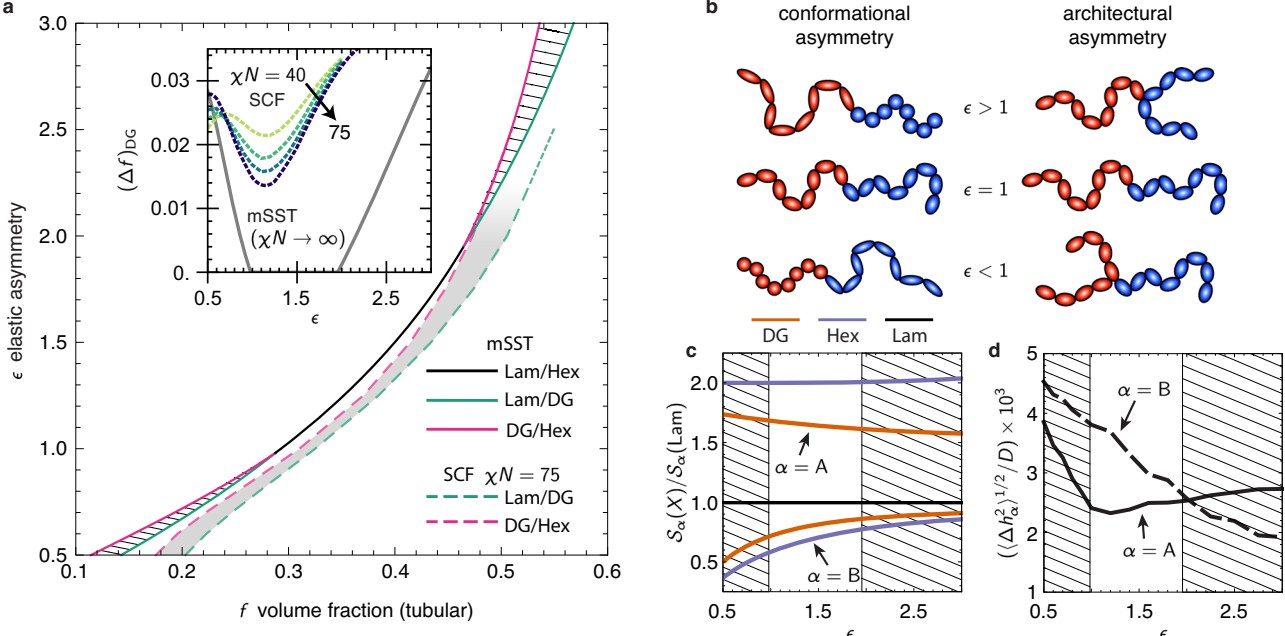

**Fig. 4 Effects of elastic asymmetry on double-gyroid stability. a** Shows the stability of DG, Hex and Lam in the $f$-$\epsilon$ plane for medial SST (mSST), with hatched regions indicating stable DG windows. Dashed curves indicate corresponding boundaries for the SCF at $\chi N = 75$, and the inset compares the composition width of the DG stability region $(\Delta f)_{DG}$ between infinite- and finite-segregation theories, including results for $\chi N = 40$, 50, and 60. **b** Schematic illustration of the dependence of *elastic asymmetry* on conformational (segment length) asymmetry and architectural (branch) asymmetry for AB miktoarm copolymers. **c** compares the block stretching costs between the competitor morphologies along the $F(\text{Lam}) = F(\text{Hex})$ line for variable elastic asymmetry, while (**d**) compares the variance in domain thickness in DG domains along the same line. In both, the hatched regions shown predicted DG windows.

derives from the reduced stretching in the tubular block, consistent with the reduced stretching of A blocks near point **K** defined in Fig. 1d, e. The entropic gain due to medial packing brings DG into near-degeneracy (to within free energy 0.02% excess) with its competitors at the boundary between Lam and Hex at $f \simeq 0.29$. Next, we consider how elastic asymmetry between the domains lifts that degeneracy.

**Elastic asymmetry and double-gyroid stability.** Beyond the volume fraction $f$ of tubular blocks, stability of block copolymer phases is known to be critically dependent on chain architecture and conformational rigidity. The *elastic asymmetry* parameter $\epsilon \equiv (n_B a_A)/(n_A a_B)$, as defined by Milner for $A_{n_A}B_{n_B}$ miktoarm stars[43] and illustrated in Fig. 4b, encodes the relative stiffness between blocks of unequal segment length $a_A \neq a_B$ (conformational asymmetry) and number of branches $n_A \neq n_B$ (architectural asymmetry). In terms of Eq. (1), $\kappa_B/\kappa_A \propto \epsilon$, so that when $\epsilon > 1$, the B matrix domains are stiffer compared to the A tubular domains[16,42]. Notably, elastic asymmetry in sphere phases has been understood to unlock a host of "self-alloying" Frank–Kasper crystal structures through the amplification of packing frustration in the coronal blocks[16,44–46].

Figure 4a shows the phase boundaries between Lam, Hex and DG based on the medial SST, in the $f$-$\epsilon$ plane. Surprisingly, the thermodynamic stability of DG exhibits a non-monotonic dependence on elastic asymmetry. For intermediate values of asymmetry $0.98 < \epsilon < 1.96$ there is no stable DG phase, and only a line of direct order-order transitions between Hex and Lam. However, for sufficiently high or low values of asymmetry (i.e. $\epsilon \geq 1.95$ and $\epsilon \leq 0.98$) stable windows of DG open up intermediate to Hex and Lam (SST based on skeletal ansatz[26] showed previously that DG becomes stable for the especially large elastic asymmetry $\epsilon \gtrsim 9.5$.).

For comparison, we show in Fig. 4a phase boundaries at finite segregation ($\chi N = 75$) from SCF calculations[47]. Aside from a slight offset to larger $f$ (attributed to finite-$\chi N$ effects), the skewing of phase boundaries to larger $f$ with increasing $\epsilon$ agrees with medial SST predictions. Notably, the width $(\Delta f)_{DG}$ of stable DG compositions exhibits a non-monotonic dependence on $\epsilon$, widening for large and small asymmetry values, as shown in the inset of Fig. 4a. This suggests that the coupling between elastic asymmetry and packing frustration captured by medial SST persists even at finite $\chi N$.

We consider the mechanism underlying the non-trivial dependence of DG on elastic asymmetry, based on the dimensionless entropic costs $S_\alpha(X)$ of competing phases plotted along the line $F(\text{Lam}) = F(\text{Hex})$ for variable $\epsilon$ in Fig. 4c. The stability of DG for $\epsilon > 1.96$ is consistent with the long-held notion that packing in the tubular block is problematic[15], so that discounting its thermodynamic cost relative to the matrix should increase its stability. There is a more subtle scenario in the Hex phase, where the coronal domain is increasingly frustrated with increasing $\epsilon$, causing the IMDS to warp hexagonally[42], which in turn deforms the core domain, as shown in the relative increase of entropic costs of the A block with $\epsilon$ (see Supplementary Note 6). In contrast, terminal spreading in the tubular block of DG leads to a relative decrease of those entropic costs with $\epsilon$, allowing this phase to overtake Hex for sufficiently stiff matrices.

The "reentrant" stability of DG for $\epsilon \lesssim 0.98$ is more confounding, as the increased cost of packing frustration in the tubular block would seemingly destabilize the DG, according to the prevailing notion that suggests packing frustration is concentrated within the tubular block[15]. Instead we observe that the favorable A block stretching of DG relative to Hex is maintained and the nominal thickness of the tubular domain shrinks with composition along the $F(\text{Lam}) = F(\text{Hex})$ line. This suggests the optimal DG morphology is able to redistribute the

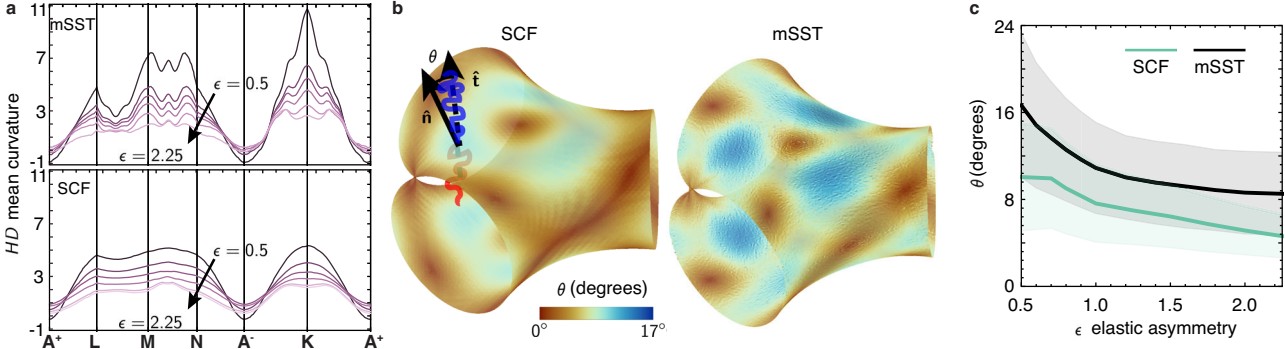

**Fig. 5 Finite- versus infinite-segregation models of domain shape and packing. a** Band diagrams showing local mean curvature $H$ along the path defined in Fig. 1d of the IMDS of a DG calculated via medial SST and SCF. **b** Maps of chain tilt at the IMDS for SCF and medial SST (mSST) at $\epsilon = 1$ and $f = 0.3$ with (**c**) the corresponding dependence of tilt-angle distributions for DG morphologies along the $F(\mathrm{Lam}) = F(\mathrm{Hex})$ line (center line gives the mean value and range shows standard deviation).

cost of packing frustration from the tubular domain to the matrix domain, in which thermodynamic costs are relatively less costly for $\epsilon < 1$. This is consistent with comparisons of the r.m.s. variance of block thickness in the A and B domains, plotted in Fig. 4d. While DG exhibits a lower thickness variation in the matrix blocks for $\epsilon \gtrsim 2$, as $\epsilon$ is decreased, the optimal DG morphology reorganizes to reduce the thickness variation in stiffer, tubular domain. Taken together, medial SST predicts a nuanced picture of packing frustration in DG, in which terminal spreading along medial sets facilitates a remarkable degree of adaptability to a broad range of molecular structure.

**Fingerprints of medial packing at finite segregation.** Medial SST predicts that optimal arrangements of DG assemblies rely on a variable spreading of chain termini along medial sets interior to the domains. Here, we consider morphological features of DG assemblies and their variation with elastic asymmetry that derive from medial packing in the $\chi N \to \infty$ limit, but persist at finite segregation. In Figure 5 we compare DG morphologies modeled by SCF at $\chi N = 75$ along the points $(\epsilon, f)$ where $F(\mathrm{Lam}) = F(\mathrm{Hex})$ according to medial SST calculations. Notably, SCF calculations include chain fluctuation effects associated with finite IMDS widths and contact zones between opposing brushes, without a priori assumptions about the underlying chain configurations. In Fig. 5a, we compare "band diagrams" of the spatial map of IMDS mean curvature $H$ (along the path introduced in Fig. 1d) for a range of elastic asymmetry, $0.5 < \epsilon < 2.25$. These show qualitatively similar spatial patterns between medial SST and SCF, with characteristically flatter regions along the threefold directions (A$^{\pm}$) and maximal curvature at the elbows (**K**), although magnitudes and fine features differ between infinite- and finite-segregation models (analogous comparisons are shown for Gaussian curvature in Supplementary Fig. 10A, B). More significantly, the variance of mean curvature is shown to decrease systematically with increasing elastic asymmetry (see Supplementary Fig. 10C), suggesting that IMDS shapes tend more towards area-minimizing[14,20] (i.e. constant mean curvature) shapes as the ratio of matrix to tubular block stiffness grows.

Underlying the adaptation of the DG geometry is the increasing lateral spread of the terminal webs of the tubular blocks with increasing elastic asymmetry, shown in Supplementary Fig. 6. This trend might suggest a discernible measure of displacement of free A block ends from the skeletal graph for increasing $\epsilon$. In Supplementary Fig. 14, we compare the distance distributions of free A block ends from the skeletal graphs for three values of $\epsilon$. While there is indeed a systematic shift of end distributions away from the skeleton observed for increasing $\epsilon$ in

both SCF and SST results, free ends are always well-distributed throughout brushes that any signal of medial spreading in chain ends is rather diffuse.

For an alternative molecular fingerprint of the underlying terminal boundaries, we turn to chain orientation. As noted above, in the medial SST construction, local volume balance extending away from the same points on the IMDS requires some measure of chain tilting from the local IMDS normal, a signature of deviation from strictly medial packing. Using the polar order parameter extracted from SCF[48] and the direct trajectories from optimal medial SST configurations, we compare the map of the local tilt angle $\theta$ relative to the IMDS at finite-$\chi N$ and $\chi N \to \infty$ in Fig. 5b. Both show strikingly similar spatial patterns of tilted and normal regions. Although finite-$\chi N$ results show a smaller magnitude of tilt, both SCF and medial SST show near-identical patterns of normal packing ($\theta = 0$). This occurs for the quasi-planar threefold points (A$^{\pm}$) along the elbow (extending from **N** to **K**) and also at a satellite spot along the strut (between **L** and **M**). More careful analysis (Supplementary Fig. 13) shows that this satellite spot is located precisely along the ray that is co-normal to the underlying terminal web (a line of 2-fold symmetry), and hence the rotation of that normal zone reflects (half of) the 70.5° twist between nodes in DG. In this way, we observe the non-trivial pattern of tilt at the IMDS as an indirect image of the terminal packing within the domain.

Figure 5c shows the magnitude and variation of the tilt distribution as function of elastic asymmetry. While magnitudes of tilt are larger for infinite- compared to finite-segregation theories, both show a consistent trend decreasing from relatively high tilt for $\epsilon < 1$ to approach a more normal (i.e. $\theta \to 0$) packing for $\epsilon > 1$. This latter tendency towards normal orientation implies that chain packing approaches closer to the medial limit as matrix domains stiffen and stable DG morphologies shift to larger A-block compositions, ultimately, becoming majority tubular structures for $\epsilon \gtrsim 2.25$.

## Discussion

In summary we have developed an approach to the SST limit of AB block copolymer melts which constructs space-filling packings of chains from medial sets, and applied it to reassess the thermodynamic stability of the DG phase. In contrast to the prior packing ansatz[26,28], which assumed that chain termini are localized to the 1D skeletons within the tubular networks, we consider molecular configurations that span from medial sets of DG generating surfaces, accounting for a substantially lower entropic cost for occupying the complex domain shapes at constant density. Our variational framework considers, explicitly, the free

energy dependence of DG morphologies as terminal ends in the tubular blocks vary their degree of localization towards or away from the 1D skeleton, with optimal structures exhibiting a broad degree of spreading over twisted web-like surfaces through tubular domains. This lower packing frustration cost qualitatively alters the thermodynamic picture for DG stability, and notably its dependence on elastic asymmetry. The SST based on skeletal packing predicts that DG only becomes increasingly stable relative to its Hex and Lam competitors as the matrix block is stiffened, and ultimately only becomes an equilibrium phase at experimentally exotic conditions, for $\epsilon \gtrsim 9$ (the equivalent of a $AB_9$ miktoarm star of equal A and B segment lengths)[26]. In contrast, the lower free energy achieved by medial packing implies that DG becomes an equilibrium phase under the much more typical experimental range of $\epsilon \approx 1$. Moreover, the thermodynamics of the DG is shown to have a far more complex, non-monotonic dependence on elastic asymmetry, counterintuitively increasing in stability as the tube-forming block becomes relatively stiff, as well as in the opposite regime of stiffer matrix.

Notably, our predictions suggest that complex networks, with suitably low elastic asymmetry, can be formed in equilibrium down to surprisingly low tubular volume fractions (at least as low as 15%). In addition to confirmation by SCF computations at even higher segregation strength, the stability of these "narrow tube" networks, as well as the strong dependence of the stability window in the range $0.5 \lesssim \epsilon \lesssim 2.5$, should be experimentally accessible via carefully adjusted combinations of branch and segment length asymmetry, along the lines of recent experiments on the effects of elastic asymmetry on complex sphere phase formation[49,50]. These thermodynamic predictions are predicated on a revised picture of the underlying subdomain packing of chains. We have highlighted how the medial packing motif impacts detailed distributions of IMDS curvature (Fig. 5) and their sharp contrast with skeletal packing motifs Supplementary Fig. 11. Combined with recent advances in spatial mapping of subdomain IMDS shape via 3D tomography (i.e. slice and view scanning electron microscopy)[51], these curvature distributions and their variation with elastic asymmetry and composition would provide at least an indirect fingerprint of medial packing. More direct confirmation of the detailed subdomain packing distribution requires reconstruction of the terminal geometry itself, or if not, spatial maps of the area per chain at the IMDS, both of which motivate experimental approaches to characterize 3D geometry of specifically labeled subregions of complex morphology[20].

Our results establish the relevance of medial geometry to the molecular degrees of freedom of network assembly of block copolymer melts. While constraints of local volume balance in this neat system frustrate perfect medial packing (i.e. normal to IMDS) of chains, variable spreading of terminal ends of blocks along the medial sets is nonetheless essential to the stability of DG in the $\chi N \rightarrow \infty$ limit. Moreover, we observe that the degree of conformity with perfect medial packing for DG itself varies with molecular parameters controlling the ratio of domain stiffnesses. In this light, the medial SST reformulates the notion of packing frustration for block copolymer assembly, and we anticipate implications well beyond the stability of DG in melt assembly.

First, the medial SST approach extends straightforwardly to other network morphologies, such as the cubic cousins of DG—DD and DP. Indeed, it can be shown that medial SST variants of DD and DP exhibit lower free energies than the prior skeletal ansatz. However, over the range of conditions presented here for AB block copolymer melts, the free energies of DD and DP always exceed that of DG. As we will describe elsewhere, we expect that the stability of DG over these competitors derives

largely from the uniquely smooth geometry of the terminal webs of DG. Notably, there is reason to suspect from SCF calculations that equilibrium stability of cubic networks (i.e. DG) could give way to stable "layer networks" in the form of perforated lamellar morphologies at particularly low values of elastic asymmetry in AB mitkoarms[52] or more complex architectures[53], although the physical mechanism for such a transition is not yet clear.

Beyond network phases, it can be expected that medial packing motifs play a key role in other complex morphologies, notably the Frank–Kasper (FK) phases of sphere-like domains that are stabilized at high elastic asymmetry[46]. It is well appreciated that phases "self-alloy" into multiple symmetry- and volume-inequivalent domains. Crudely speaking, the terminal boundaries between coronal blocks are polyhedral cells, similar to Voronoi cells. However, medial analysis of FK-forming domains suggests that the terminal packing in core domains is more subtle[20]. These FK structures (e.g. A15 and C15) are composed of populations of more isotropic domains, with inner medial sets localized around points, mixed with another population of more anisotropic domains possessing planar, disk-like medial sets. Prior SST models for explaining FK phase formation are based on the evidently oversimplified assumption that core termini are localized to the centroids of domains[16,45], and hence likely considerably overestimate the entropic costs in these anisotropic domains, whose local packings include quasi-lamellar regions. We anticipate that properly accounting for terminal spreading via an extension of medial SST may be critical for understanding open questions in FK formation. For example, what controls the sequence of stable sphere phases (BCC $\rightarrow \sigma \rightarrow$ A15) found when increasing elastic asymmetry and core block fraction[44,46]? Additionally, end-exclusion zone corrections to parabolic brush theory, while seemingly unimportant for the network phases (Supplementary Note 10), are expected to yield substantial free energy corrections for sphere phases[41] and may therefore have a significant effect on predicting which sphere phases are stable.

Lastly, we briefly comment on likely effects of medial packing beyond neat (i.e. melt) systems. In blends, adding low-molecular weight species that dissolve in either one or both blocks (i.e. the so-called wet brush regime) will certainly have the effect of introducing compressibility to those solvated domains. In the context of medial packing specifically, compressibility has the effect of relaxing the local volume balance constraint, and thereby should reduce the frustration of medial packing. Hence, we speculate that a direct consequence of blending with low-molecular weight solvent species should be a progression toward un-tilted (i.e. medial) packing, which may establish a convenient conceptual link between medial packing frustration in neat (thermotropic) and concentrated (lyotropic) phases. Mixtures of block copolymers with high-molecular weight species (e.g. homopolymer of A or B type) are generally expected to lead to local segregation of additives from the molten brushes comprising each domain[15,54], in effect sequestering them into interstitial "hot spots" between those brushes, in turn relaxing the free energy cost of filling space for those local regions. SCF results[54–56], and experiments[57], show that the stable network morphology can be altered upon blending homopolymer to the tubular domain, from DG to DD, and potentially to DP upon further increased homopolymer fraction. Analogous effects altering the symmetry of the stable morphology have been observed and studied theoretically in FK-forming sphere phases of block copolymer and homopolymer blends[58–60]. While it is arguably intuitive that homopolymer blends relax packing frustration in complex phases, it remains to be understood what geometric features of these morphologies lead a complex "host" to gain or lose stability over another upon loading with a "guest" species. We anticipate that hot spots can themselves be directly

related to the medial geometry of a complex morphology, and moreover, the medial SST construction may be readily extended to provide a direct map of the location and thermodynamics of these hot spots upon blending.

## Methods

**Medial strong segregation theory for double gyroid**. The medial strong segregation theory calculation optimizes the SST free energy, Supplementary Eq. (11), over the variational family of generating gyroidal surfaces to determine and analyze thermodynamically preferred DG chain packing. Complete details on the construction of tessellating chain configurations and the variational algorithm are given in Supplementary Note 3–4.

We begin with a broad set of generating surfaces **G** with $Ia\bar{3}d$ symmetry; here we have chosen **G** as level sets of a three Fourier-mode expansion of the gyroid symmetry. From the medial map of **G** (for the double network) we generate the terminal boundaries of both domains, $T_A$ and $T_B$, referring to the tubular and matrix domains respectively. The induced map between $T_A$ and $T_B$ approximates the mean trajectories of domain-spanning chains, which occupy local volumes approximated by triangular prismatic wedges, thus providing a complete tessellation of the DG structure. The location of the IMDS is then determined along each trajectory according to the constraints of volume balance and chain composition, resulting in mutually compatible terminal boundaries, IMDS, and trajectories, as shown in Fig. 2b, from which we compute the geometric contribution to the free energy in Eq. (1). Notably, local volume balance generally requires some measure of tilt between trajectories and the IMDS, such that the terminal surfaces of the ultimate volume-balanced packing deviate at least slightly from the medial surfaces of the final IMDS. As shown in Fig. 2c, variation of the generating surface leads to variation in compatible IMDS shapes and variable spreading of the terminal boundaries for fixed block composition *f*, which we then optimize over (see Supplementary Note 4).

**Skeletal strong segregation theory ansatz for double gyroid**. We employ a variant of the medial SST approach to model the DG assemblies whose terminal sets are constrained to lie along the skeletal graphs of the double network, in a effort to closely recapitulate and compare to prior results from ref. [26]. Complete details are provided in Supplementary Note 5. In brief, we construct a variational set of skeletal tessellations via projection of the terminal positions in the tubular domains from the "web-like" medial surfaces of double-gyroids to the nearby positions along the skeletal graph. Finally, we optimize the free energy over the free parameters of the map that takes points on the tubular domain medial surface to points on the skeletal graph.

**Strong segregation theory of the hexagonal columnar phase**. To compute the SST free energy of the competitor Hex phase we use the method of ref. [42], which considers a variable class of IMDS shapes—intermediate to circular and hexagonal shapes—and corresponding space-filling (so-called kinked-path) chain trajectories chain trajectories within domains. Complete details and subdomain packing analysis provided in Supplementary Note 6.

**Finite segregation strength self-consistent field calculations**. Self-consistent field (SCF) calculations of Lam, Hex and DG phases are done using the PSCF code as described in ref [47]. Distributions of the free end junctions are computed from the SCF segment distribution functions as detailed in Supplementary Note 9. As detailed in Supplementary Note 8, the tilt of the chain trajectories is computed from the (polar) order parameter at the IMDS (i.e. regions composed equally of equal parts A and B segments), which is computed from spatial gradients of segment distribution functions as described in ref. [48].

**IDMS shape analysis**. For the IMDS shape analysis, we adapted methods for calculating smooth approximations to curvature on a discrete meshes based on least-squares fitting of well-behaved quadric surfaces to small surface patches. The quadric surface fits provide for smooth evaluations of curvature that are local to each surface patch. We obtain a global curvature distribution by stitching together the collection of surface patches, averaging over individual patch curvature evaluations at the intersection of multiple patches. This algorithm also provides a route to smoothing over rough meshes, such as those that appear in the skeletal SST construction. This algorithm is described in detail in Supplementary Note 7A. To construct the "band diagrams," we specify a set of cutting planes and find their intersections with a given meshed surface, yielding curved patches on the surface. We then identify the mesh edges that intersect each plane and use linear interpolation to evaluate vertex- or face-addressed data at the location where each edge intersects with a plane. This method is outlined in Supplementary Note 7B.

**End-exclusion zone corrected free energies**. End-exclusion zone (EEZ) corrections to the parabolic brush theory calculations of chain stretching free energy, which are described in Supplementary Note 10, were found using the results of[41].

We incorporated a look-up table and interpolation tool available in the supplementary code of that reference (available at https://doi.org/10.7275/rtvx-h237) to adjust the stretching free energy for each wedge in our medial SST calculations.

## Data availability

Computational results are available at https://doi.org/10.7275/s98r-sb17.

## Code availability

Supporting software codes for medial SST model of DG are available at https://doi.org/10.7275/s98r-sb17.

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

## Acknowledgements

We are grateful to E. Thomas, G. Schröder-Turk, F. Bates, K. Dorfman and S. Park for stimulating discussions and valuable comments on this work. This research was supported by the US Department of Energy (DOE), Office of Basic Energy Sciences, Division of Materials Sciences and Engineering under awards DE-SC0014599 and DE-SC0022229. Medial SST and SCF calculations were performed on the UMass Cluster at the Massachusetts Green High Performance Computing Center.

## Author contributions

A.R., M.D., and G.G. designed and performed research and wrote the paper.

## Competing interests

The authors declare no competing interests.
