## [Peer Review File · Nature Communications]

Medial packing and elastic asymmetry stabilize the double-gyroid in block copolymersREVIEWER COMMENTS

Reviewer #1 (Remarks to the Author):

In this manuscript the authors carried out a detailed study on the relative stability of the double-gyroid (DG) structure self-assembled from diblock copolymers. Specifically, the authors developed a much improved variational ansatz, i.e. the medial packing model, for the different ordered phases within the framework of strong-segregation theory, and used the model to show that the phase could be stabilized by varying the volume fraction and the conformational asymmetry of the block copolymers. These are new and insightful results for a long-standing problem in soft matter physics, i.e., understanding the origin of the complex ordered phases. The results from the medial packing model are compared with that from the numerical self-consistent field theory (SCFT) with good agreement. It has been argued heuristically that packing frustration is the origin of the complex ordered phases (DG, Frank-Kasper phases, etc.) of block copolymers. However, how to quantify frustration so that this intuitively attractive concept could be used to predict which phase would prevail is an unsolved problem. The model and method proposed by the authors, as reported in the current manuscript, could be considered as a first step in developing a framework to quantify packing frustration in strongly-segregated block copolymers. Taking together, the model, methods and results reported in the current manuscript represents some noteworthy results on the understanding of the origin of bicontinuous phases of self-assembled from amphiphilic molecules. The concepts and principles could be generalized to other ordered phases. The paper is well written and the SI gives a detailed description of the model and method. Based on these observations, I would recommend the publication of the manuscript.

There are a few comments for the authors to consider when they revise their manuscript.

1. Fundamentally, the medial packing model is an improved variational model for the structure. This point is implied in the manuscript. It might be better to state this explicitly.
2. On page 11, the "Fig. 4B" in the sentence "For comparison, we show in Fig. 4B ..." should be "Fig. 4A".
3. At the top of page 11, the authors mentioned that increasing the conformational asymmetry would lead to a large deformation of the IMDS of the hexagonally packed cylinders. Would this lead to the formation of "complex" 2D phases, similar to the Frank-Kasper phases in 3D? It would be interesting to consider and comment on this possibility.
4. The "band diagram" (Fig. 5A) show that the SST results has more features than that from the SCFT results. Why this is so? Naively, I would expect more features from SCFT since it is a more "complete" theory.
5. As mentioned by the authors in the final paragraph, block copolymer blends provide an efficient route to obtain bicontinuous phases and FK phases. One example is that SCFT calculations by Lai and Shi (Macromolecular Theory and Simulation 30, 2100019 (2021)) showed that adding homopolymers-like diblock copolymers could stabilize the DD and DP phases. Similar SCFT calculations (Giant 5, 100043 (2021); MTS 2021, 2100053) showed that the addition of A or AB could stabilize the Frank-Kasper phases. Extending the medial packing model to these blending systems would be very interesting.

Reviewer #2 (Remarks to the Author):

In the first instance I would like to point out that I do NOT consider myself to be an expert in the chemistry/physics of block copolymers. Though I am aware of these systems and have a general interest in packing problems and self assembly. Hence, I cannot provide an evaluation of how important or ground-breaking these result may or may-not be.

However, as a non-expert reader I feel that the present article is written in a very technical manner and as such somewhat "indigestible".

Firstly, there is a heavy use of abbreviations thought (IDMS, TPN, BCP, SST, mDg, sDg, DD, DP, DG...) which makes it very difficult to understand what is being discussed. I suggest that either the authors remind readers periodically what the meaning of these abbreviations are - or - if the wish to maintain the current style to provide a key somewhere listing abbreviations and their meanings.

Secondly, (again with the non-technical reader in mind) I would like to see a simple explanation of what they actually did. I have tried to read the paper several times and my impression is the following: the authors start with the gyroid structure, from the surface of the gyroid they construct a Voronoi surface and this leads to the medial map? Ok, then what? As you can see I am still having difficulty piecing this together.

Thirdly, the authors claim that "reconciling a long-standing contradiction between infinite- and finite-segregation theories...", once again if there are important results in this article, it is not obvious where they are in the mass of technical discussion or at least need to be highlighted in such a way as to be obvious to the non-expert reader.

Reviewer #3 (Remarks to the Author):

Although the DG morphology is formed by diverse self-assembly molecules, universal explanations for its stability relative to its competitors remain elusive. The authors have developed a medial strong segregation theory (mSST) and applied it to analyze the stability of double-gyroid (DG) morphology, one of triply-periodic networks (TPNs). They show that "medial packing" is essential for the stability of DG in strongly-segregated block copolymer melts. This new mSST approach successfully reconcile a long-standing contradiction between infinite- and finite-segregation theories. The results obtained by the mSST approach provide insights into the formation mechanism of the DG morphology. In particular, this SST approach predicts that the composition window of stable DG widens both for large and small elastic asymmetry, which is corroborated by self-consistent field theory (SCFT). Therefore, this beautiful approach developed in this manuscript should be helpful for the field. This manuscript is well written. I would like to recommend it to be accepted after the authors consider the following comments.

1. In Figure 4, the nonmonotonic change of the stability region of DG, especially the disappearing in the range of $0.98 < \epsilon < 1.95$, is very interesting. Moreover, the lower critical conformational asymmetry of $\epsilon = 0.98$ is very small. That such small conformational asymmetry causes the DG phase to disappear seems questionable. To the best of my knowledge, SCFT does not predict the absence of the DG phase in AB diblock copolymer melts even with conformational asymmetry. The results of SCFT with $\chi N = 75$ shows a nonmonotonic change of the width of the DG region. It may be helpful to show the changing trend of the width of the DG region with χN for a value of ϵ in the region of $0.98 < \epsilon < 1.95$ to see whether there is possibility for the DG phase to disappear. If no, this disagreement should result from the approximation used in the mSST approach. Then, the authors could provide some explanations.
2. The authors mention that the mSST approach can extend straightforwardly to other morphologies. I don't doubt this. Another question I am concerned about is whether this method can be extended to other block copolymers, e.g. multiblock copolymers, because a lot of interesting self-assembly behaviors have been predicted in purposely designed multiblock copolymers. For example, the DG phase is replaced by the perforated-lamella (PL) phase at the high segregation region in the phase diagram of a nonlinear AB-type block copolymer (Macromolecules 2018, 51, 1529-1538); The region of the DG phase changes nonmonotonically with the architectural parameter in (BAB)_n star copolymer melts (Macromolecules 2018, 51, 9890-9900), which is similar to or different from the mechanism discussed in the current work? The region of the DG phase even changes in a more complicated manner in A(AB)_n melts (Macromolecules 2020, 53, 10907-10917).
3. If the mSST approach can extend to the Frank-Kasper phases formed by nonlinear block copolymers, it would be very useful because the transitions between the different spherical phases

become very complicated in some extremely asymmetric phase diagrams (ACS Macro Lett. 2020, 9, 668-673. Moreover, the presence of the DP phase in this reference also needs further explanations.

4. Minor points: the labels A and B are incorrect in Figure 4. A comma is missing in the sentence of "the double-gyroid (DG) double-diamond (DD) ..." on page 2.

Response to Reviewer 1:

In this manuscript the authors carried out a detailed study on the relative stability of the double-gyroid (DG) structure self-assembled from diblock copolymers. Specifically, the authors developed a much improved variational ansatz, i.e. the medial packing model, for the different ordered phases within the framework of strong-segregation theory, and used the model to show that the phase could be stabilized by varying the volume fraction and the conformational asymmetry of the block copolymers. These are new and insightful results for a long-standing problem in soft matter physics, i.e., understanding the origin of the complex ordered phases. The results from the medial packing model are compared with that from the numerical self-consistent field theory (SCFT) with good agreement. It has been argued heuristically that packing frustration is the origin of the complex ordered phases (DG, Frank-Kasper phases, etc.) of block copolymers. However, how to quantify frustration so that this intuitively attractive concept could be used to predict which phase would prevail is an unsolved problem. The model and method proposed by the authors, as reported in the current manuscript, could be considered as a first step in developing a framework to quantify packing frustration in strongly-segregated block copolymers. Taking together, the model, methods and results reported in the current manuscript represents some noteworthy results on the understanding of the origin of bicontinuous phases of self-assembled from amphiphilic molecules. The concepts and principles could be generalized to other ordered phases. The paper is well written and the SI gives a detailed description of the model and method. Based on these observations, I would recommend the publication of the manuscript.

We are grateful for the reviewer's positive and enthusiastic assessment of our work. We are especially pleased that they share our vision for how our approach, and its conceptual underpinnings, open the door to address a number of critical, unanswered questions about how molecular assemblies of block copolymers "choose" between different complex self-assembled morphologies in a more precise and predictive way.

We are also grateful for the number of suggestions and comments raised by this reviewer, and address these below, and in our manuscript, where appropriate.

There are a few comments for the authors to consider when they revise their manuscript.

1. Fundamentally, the medial packing model is an improved variational model for the structure. This point is implied in the manuscript. It might be better to state this explicitly.

We have incorporated this language into our revised discussion (the first paragraph) as well as revised introduction to our method in Sec. II of the manuscript.

2. On page 11, the "Fig. 4B" in the sentence "For comparison, we show in Fig. 4B ..." should be "Fig. 4A".

We thank the reviewer for noticing this error and have corrected it and a few other mis-labeled figure references.

3. At the top of page 11, the authors mentioned that increasing the conformational asymmetry would lead to a large deformation of the IMDS of the hexagonally packed cylinders. Would this lead to the formation of "complex" 2D phases, similar to the Frank-Kasper phases in 3D? It would be interesting to consider and comment on this possibility.

As we understand it, the reviewer raises an interesting question about whether the polyhedral "faceting" of the Hex phase might lead to symmetry-breaking transitions to more complex columnar phases, which would be the analogy to what is understood for the 3D sphere phases that transition from simple (BCC) to complex (Frank Kasper) phases when IMDSs become warped. We certainly have not investigated possibility in detail as it is rather beyond the focus of the present study, and the particular issue of the packing frustration and IMDS warping of Hex phases have received previous study (e.g. <https://doi.org/10.1021/ma049255d>). Here, we only briefly note that, as described in <https://doi.org/10.1016/j.physrep.2006.08.001>, there may be less reason to suspect transition of this type for 2D phases. Unlike the 3D case, the Hex packing is optimal (at least among 2D lattices) for both geometric stretching and area moments that couple to entropic and enthalpic free energy contributions. That said, we are not aware that a careful study of "alloyed" 2D packings of unequal volumes has been carried out to more directly address the possibility. At this point, while we agree that the issue deserves some more careful consideration, our understanding of this more complex possibility is too speculative to be of use to readers, and commenting on it further would likely distract from the core focus on the network morphologies.

4. The "band diagram" (Fig. 5A) show that the SST results has more features than that from the SCFT results. Why this is so? Naively, I would expect more features from SCFT since it is a more "complete" theory.

This is an interesting observation. We believe that the features shown in the SST "band diagram" may be residual artifacts of the few-mode representation of the IMDS that we use to keep the number of variational parameters small. On the other hand, the SCFT calculations use a much finer resolution (i.e. a high-mode representation), so any artifacts would be moved to smaller length scales. Since the algorithm we use for calculating curvature involves fitting surface patches to small regions of the surface, such small length scale artifacts would likely not be detected. Moreover, as SCFT calculations were performed at finite (and relatively modest) χN , they incorporate effects that represent sub-leading corrections to SST, which are not guaranteed to be small. We would in general expect finite χN corrections to somewhat smoothen the IMDS features relative to the asymptotic limit corresponding to SST. We would

expect better agreement between SST and SCFT if (i) the SCFT calculations were performed at higher χN and (ii) more variational modes were used in the SST calculation.

5. As mentioned by the authors in the final paragraph, block copolymer blends provide an efficient route to obtain bicontinuous phases and FK phases. One example is that SCFT calculations by Lai and Shi (*Macromolecular Theory and Simulation* 30, 2100019 (2021)) showed that adding homopolymers-like diblock copolymers could stabilize the DD and DP phases. Similar SCFT calculations (*Giant* 5, 100043 (2021); *MTS* 2021, 2100053) showed that the addition of A or AB could stabilize the Frank-Kasper phases. Extending the medial packing model to these blending systems would be very interesting.

We thank the reviewer for pointing out the additional references on blending diblocks and homopolymers which we have incorporated into our revised manuscript.

Response to Reviewer 2:

In the first instance I would like to point out that I do NOT consider myself to be an expert in the chemistry/physics of block copolymers. Though I am aware of these systems and have a general interest in packing problems and self assembly. Hence, I cannot provide an evaluation of how important or ground-breaking these result may or may-not be.

However, as a non-expert reader I feel that the present article is written in a very technical manner and as such somewhat "indigestible".

We are grateful for the critical perspective of Rev. 2, and concede that many aspects of our manuscript were not sufficiently accessible for an audience not already well versed in certain aspects of block copolymer physics. In this work, we exploit theoretical tools for modeling phase segregated domains of block copolymers to specifically test the link between molecular configurations, medial geometry and thermodynamics of network self assembly. That is, we focus on the case of block copolymers where we can make the geometric notions and thermodynamic consequences of packing frustration specific and quantitative. That said, we certainly expect that many of the key lessons of our study will have analogous implications for other molecular systems (e.g. volume balance and density constraints frustrates perfect "medial" packing, leading to complex tilt patterns), although the theoretical formalisms to quantify these will have to be different. Aspects of this medial packing hypothesis have previously been raised for other amphiphile (namely, lipid) assemblies (<https://doi.org/10.1140/epjb/e2003-00308-y> and <https://doi.org/10.1039/C2FD20112G>). In that light, we certainly hope that our manuscript is more "digestible" for that non-block copolymer physics audience. Following the suggestions of this reviewer and the editor, we have revised the manuscript to improve its readability with this broader group of readers in mind.

Firstly, there is a heavy use of abbreviations throughout (IDMS, TPN, BCP, SST, mDg, sDg, DD, DP, DG...) which makes it very difficult to understand what is being discussed. I suggest that either the authors remind readers periodically what the meaning of these abbreviations are - or - if the wish to maintain the current style to provide a key somewhere listing abbreviations and their meanings.

We thank the reviewer for this very helpful suggestion, which we have opted to follow in our revised manuscript. We have eliminated the use of BCP, TPN, mSST, mDG, and sDG, replacing these with unabbreviated versions. We agree that this improves the overall readability of the text.

Secondly, (again with the non-technical reader in mind) I would like to see a simple explanation of what they actually did. I have tried to read the paper several times and my impression is the following: the authors start with the gyroid structure, from the surface of the gyroid they construct a Voronoi surface and this leads to the medial map? Ok, then what? As you can see I am still having difficulty piecing this together.

We appreciate that the reviewer is asking for a more approachable synopsis for “what was done”. We have addressed this in two ways. First, we have added a 2 sentence “bullet summary” of the core algorithm underlying the medial strong-segregation construction in the 3rd paragraph of Sec. II. There, we reference a new figure in the SI (Fig. S7), which graphically summarizes the “workflow” of our algorithm, in order to give the reader a quick overview of how the construction and calculation works without requiring a detailed reading of the methods. Second, we have revised our discussion to begin in the first paragraph with a “top line summary” of the advance of the theoretical framework and how it was applied and what predictions were made.

Thirdly, the authors claim that "reconciling a long-standing contradiction between infinite- and finite-segregation theories...", once again if there are important results in this article, it is not obvious where they are in the mass of technical discussion or at least need to be highlighted in such a way as to be obvious to the non-expert reader.

Similar to the point raised above, the reviewer’s comment implies the need for a simple summary of “what was found?” and “why is it important?”. The added first paragraph of the discussion includes a restatement of the key findings, and how these resolve the basic puzzle that prior strong segregation theory found that double gyroids (under common experimental conditions) are unstable while best attempts to assess this stability from experiments or self-consistent field theory suggest that gyroid is an equilibrium phase in the limit of strong-segregation. Combining these additions together with recommended comments on experimental implications suggested by the editor, we feel that the revised opening to the discussion provides a more accessible synopsis of these top-level points about our work

Response to Reviewer 3:

Although the DG morphology is formed by diverse self-assembly molecules, universal explanations for its stability relative to its competitors remain elusive. The authors have developed a medial strong segregation theory (mSST) and applied it to analyze the stability of double-gyroid (DG) morphology, one of triply-periodic networks (TPNs). They show that “medial packing” is essential for the stability of DG in strongly-segregated block copolymer melts. This new mSST approach successfully reconciles a long-standing contradiction between infinite- and finite-segregation theories. The results obtained by the mSST approach provide insights into the formation mechanism of the DG morphology. In particular, this SST approach predicts that the composition window of stable DG widens both for large and small elastic asymmetry, which is corroborated by self-consistent field theory (SCFT). Therefore, this beautiful approach developed in this manuscript should be helpful for the field. This manuscript is well written. I would like to recommend it to be accepted after the authors consider the following comments.

We are grateful for the enthusiastic and positive assessment of our work, as well as for the many insightful comments raised by this reviewer, which we respond to below.

1. In Figure 4, the nonmonotonic change of the stability region of DG, especially the disappearing in the range of $0.98 < \epsilon < 1.95$, is very interesting. Moreover, the lower critical conformational asymmetry of $\epsilon = 0.98$ is very small. That such small conformational asymmetry causes the DG phase to disappear seems questionable. To the best of my knowledge, SCFT does not predict the absence of the DG phase in AB diblock copolymer melts even with conformational asymmetry. The results of SCFT with $\chi N = 75$ shows a nonmonotonic change of the width of the DG region. It may be helpful to show the changing trend of the width of the DG region with χN for a value of ϵ in the region of $0.98 < \epsilon < 1.95$ to see whether there is possibility for the DG phase to disappear. If no, this disagreement should result from the approximation used in the mSST approach. Then, the authors could provide some explanations.

The reviewer is asking about the state of the evidence from SCFT that the composition window, $(\Delta f)_{DG}$, closes off for $1 < \epsilon < 2$ at infinite χN . The short answer is that there are no results from SCFT that get close enough to the asymptotic limit of $\chi N \rightarrow \infty$ to conclude on this point. As far as published results, the highest χN reports on DG only go to $\chi N = 100$ (for conformationally asymmetric diblocks), <https://doi.org/10.1021/ma0527707>. We have performed some additional SCFT calculations, first varying the degree of segregation and variable ϵ , and these are shown in the inset of fig. 4A, suggesting a trend that indeed that as χN increases the non-monotonic variation with $(\Delta f)_{DG}$ with ϵ grows and deepens, consistent with a trend towards the medial SST predictions. For the referees benefit, for $\epsilon = 1$ along, we have pushed up SCFT solutions to $\chi N = 180$, and these suggest at this range segregation the composition is

continuing to narrow down to 1.2%. Careful analysis (DOI 10.1140/epje/i2010-10673-4) of the approach of phase boundaries between “classical” morphologies (i.e. lamella and cylinders) shown that transition values continue to shift in composition by at least this much up to $\chi N \sim 10^3 - 10^4$. Suffice it so say that there is some disagreement among experts (see discussion of this point in DOI 10.1140/epje/i2009-10534-3) about what conclusions or extrapolations might be pointing to based on data down at the the range of $\chi N \sim 10^2$. Certainly we hope that our study will stimulate efforts to push the capabilities of more advanced computational SCFT algorithms to put our predictions about the $\chi N \rightarrow \infty$ limit to the test. However, we are far from the point where we would be willing to speculate about what directions the SCFT predictions are pointing in this specific regard.

2. The authors mention that the mSST approach can extend straightforwardly to other morphologies. I don't doubt this. Another question I am concerned about is whether this method can be extended to other block copolymers, e.g. multiblock copolymers, because a lot of interesting self-assembly behaviors have been predicted in purposely designed multiblock copolymers. For example, the DG phase is replaced by the perforated-lamella (PL) phase at the high segregation region in the phase diagram of a nonlinear AB-type block copolymer (Macromolecules 2018, 51, 1529-1538); The region of the DG phase changes nonmonotonically with the architectural parameter in (BAB)_n star copolymer melts (Macromolecules 2018, 51, 9890-9900), which is similar to or different from the mechanism discussed in the current work? The region of the DG phase even changes in a more complicated manner in A(AB)_n melts (Macromolecules 2020, 53, 10907-10917).

This is an excellent question. We anticipate that “alternating” or “core/shell” morphologies of linear multiblock (e.g. ABC terblocks) copolymers could be captured by extensions of approaches suggested by ref. <https://doi.org/10.1021/ma971046o>. The questions is really about the mapping from one the terminal surface to the next, and for alternating morphologies, we expect this may require mapping between three consecutive sets, into “compound wedge” geometries that we discussed in <https://doi.org/10.1021/acs.macromol.1c00958>. More complex architectures (outside of the class A_n B_m miktoarms) can, in principle, be addressed, and the medial approach to construct and sample low free energy tessellations of complex morphologies can more or less be applied as described here. We have been investigating ways to extend mSST to incorporate multiblock copolymers and other block copolymer architectures. One simpler example, would be a miktoarm with multiple A or B block lengths, which changes the problem by giving rise to brushes of polydisperse linear blocks. More complex architectures, of the type suggested by the reviewer, would lead to brush domains that contain mixtures of linear and looped blocks, which alter stretching free energy functional for filling those domains. At present, there are no “off the shelf” extensions of strong-stretching theory as simple as the “parabolic brush” theory to address all such these cases, but there are no fundamental obstacles to addressing and studying the cases of brushes of polydisperse blocks and loops (as you would need to describe some of the more complex architectures addressed computationally in these references).

That said, we take the reviewer's additional suggestion to clarify the possibility that stable DG networks may be overtaken by other networks, even "planar" PL networks. We have included a reference to this point in our revised discussion along with the report of stable perforated lamellar windows in *Macromolecules* 2018, 51, 1529-1538.

3. If the mSST approach can extend to the Frank-Kasper phases formed by nonlinear block copolymers, it would be very useful because the transitions between the different spherical phases become very complicated in some extremely asymmetric phase diagrams (*ACS Macro Lett.* 2020, 9, 668-673. Moreover, the presence of the DP phase in this reference also needs further explanations.

At present we have so far only briefly experimented with using mSST to study spherical phases, but have largely focused on network phases (including DP). Yet we agree with the reviewer that using mSST to study chain packing in complex sphere phases would likely aid in understanding transitions between them. In particular, the sequence of multiple sphere phases encountered for extreme elastic asymmetry is still not understood from the perspective of packing frustration and crystalline symmetry. We note that SST calculations utilize results from the theory of strongly stretched brushes for the stretching free energy of individual blocks. As outlined in the SI (please refer to section S10), our construction has been adapted to account for end-exclusion zone corrections to brush theory. While seemingly unimportant for the network phases, these corrections are expected to yield substantial free energy corrections for sphere phases and *may* therefore have a significant effect on predicting which sphere phases are stable. We have included a sentence in our revised discussion to highlight this specific point.

4. Minor points: the labels A and B are incorrect in Figure 4. A comma is missing in the sentence of "the double-gyroid (DG) double-diamond (DD) ..." on page 2.

We thank the reviewer for noticing these errors, which we have now fixed.

REVIEWERS' COMMENTS

Reviewer #1 (Remarks to the Author):

In response letter and revised manuscript, the authors have addressed comments from previous reviewers satisfactorily. As such, I would recommend the publication of this manuscript in its current form.

Reviewer #2 (Remarks to the Author):

I am satisfied with the changes made by the authors. I would be happy to see the article published now.

Reviewer #3 (Remarks to the Author):

The authors have addressed my comments adequately.